

# Impact of nitrogen fertilizer type and application rate on growth, nitrate accumulation, and postharvest quality of spinach

Kemal Yalçın Gülüt and Gamze Güleç Şentürk

Department of Soil Science and Plant Nutrition/Faculty of Agriculture, Çukurova University, Sarıçam, Adana, Turkey

## ABSTRACT

**Background:** A balanced supply of nitrogen is essential for spinach, supporting both optimal growth and appropriate nitrate ($NO_3^-$) levels for improved storage quality. Thus, choosing the correct nitrogen fertilizer type and application rate is key for successful spinach cultivation. This study investigated the effects of different nitrogen (N) fertilizer type and application rates on the growth, nitrate content, and storage quality of spinach plants.

**Methods:** Four fertilizer types were applied at five N doses (25, 50, 200, and 400 mg N kg$^{-1}$) to plants grown in plastic pots at a greenhouse. The fertilizer types used in the experiment were ammonium sulphate (AS), slow-release ammonium sulphate (SRAS), calcium nitrate (CN), and yeast residue (YR). Spinach parameters like Soil Plant Analysis Development (SPAD) values (chlorophyll content), plant height, and fresh weight were measured. Nitrate content in leaves was analyzed after storage periods simulating post-harvest handling (0, 5, and 10 days).

**Results:** The application of nitrogen fertilizer significantly influenced spinach growth parameters and nitrate content. The YRx400 treatment yielded the largest leaves (10.3 ± 0.5 cm long, 5.3 ± 0.2 cm wide). SPAD values increased with higher N doses for AS, SRAS, and CN fertilizers, with AS×400 (58.1 ± 0.8) and SRAS×400 (62.0 ± 5.8) reaching the highest values. YR treatments showed a moderate SPAD increase. Fresh weight response depended on fertilizer type, N dose, and storage period. While fresh weight increased in all fertilizers till 200 mg kg$^{-1}$ dose, a decrease was observed at the highest dose for AS and CN. SRAS exhibited a more gradual increase in fresh weight with increasing nitrogen dose, without the negative impact seen at the highest dose in AS and CN. Nitrate content in spinach leaves varied by fertilizer type, dose, and storage day. CNx400 resulted in the highest $NO_3^-$ content (4,395 mg kg$^{-1}$) at harvest (Day 0), exceeding the European Union's safety limit. This level decreased over 10 days of storage but remained above the limit for CN on Days 0 and 5. SRAS and YR fertilizers generally had lower $NO_3^-$ concentrations throughout the experiment. Storage at +4 °C significantly affected $NO_3^-$ content. While levels remained relatively stable during the first 5 days, a substantial decrease was observed by Day 10 for all fertilizers and doses, providing insights into the spinach's nitrate content over a 10-day storage period.

**Conclusion:** For rapid early growth and potentially higher yields, AS may be suitable at moderate doses (200 mg kg$^{-1}$). SRAS offers a more balanced approach, promoting

Corresponding author
Kemal Yalçın Gülüt,
kygulut@cu.edu.tr

sustained growth while potentially reducing $NO_3^-$ accumulation compared to AS. Yeast residue, with its slow nitrogen release and consistently low $NO_3^-$ levels, could be a viable option for organic spinach production.

## INTRODUCTION

The global demand for leafy vegetables significantly increased in recent decades, driven by population growth and dietary shifts (*FAOSTAT, 2024*). The intensifies agricultural practices, often leading to increased use of fertilizers, a major source of nitrate ($NO_3^-$) in plants. While synthetic fertilizers are popular for their ease of application and high solubility, their excessive use can significantly elevate $NO_3^-$ levels in leafy vegetables (*Bai et al., 2021*; *Dezhangah et al., 2022*; *Luetic et al., 2023*). The increased level of $NO_3^-$ in leafy plants raises a crucial question: how can we maximize the health benefits of leafy vegetables while minimizing $NO_3^-$ and $NO_2^-$ content without compromising recommended uptake levels?

Nitrate accumulation is closely related to the nitrogen (N) metabolism of plants. Nitrogen is a crucial macro-nutrient essential for plant growth and development. Plants uptake N in the form of $NO_3^-$ and $NH_4^+$ from the soil solution. When a plant absorbs more nitrate than it can use for immediate growth and protein synthesis, an imbalance occurs. This excess nitrate can then accumulate in the plant's tissues. This phenomenon is particularly pronounced in non-leguminous crops, where higher concentrations of $NO_3^-$ tend to accumulate in the leaves, while lower levels are found in storage organs like bulbs, seeds, fruits, roots, and tubers (*Santamaria, 2006*; *Bian et al., 2020*). For this reason, leafy vegetables like spinach, lettuce, and parsley are considered prominent $NO_3^-$ accumulating species.

Several factors influence $NO_3^-$ accumulation in plants, including growing season (*Citak & Sonmez, 2010*; *M'hamdi et al., 2016*), variety, and cropping system (*Koh, Charoenprasert & Mitchell, 2012*). Studies by *Kaminishi & Kita (2006)* has highlighted the impact of seasonality on $NO_3^-$ levels in in spinach, with warmer seasons showing higher accumulation compared to colder periods Likewise, research by *Koh, Charoenprasert & Mitchell (2012)* demonstrates the influence of cropping systems, with organically grown spinach exhibiting lower $NO_3^-$ content compared to conventionally grown varieties. This difference is likely attributed to the use of synthetic fertilizers in conventional farming, leading to higher soil N availability and plant uptake. Additionally, the study by *Koh, Charoenprasert & Mitchell (2012)* suggests that spinach variety can also play a role, with some cultivars exhibiting a greater response to soil nitrogen levels than others. Supporting this notion, studies have shown that organic fertilizers release nitrogen slower than inorganic fertilizers, leading to lower $NO_3^-$ accumulation in the edible parts of organically grown crops compared to conventionally grown crops (*Colla et al., 2018*; *de Gonzalez et al., 2015*). *de Gonzalez et al. (2015)* further found that conventional vegetables contained

significantly higher $NO_3^-$ compared to organic vegetables, but there were no significant differences in nitrite content between conventional and organic vegetables. Nitrite levels generally ranged from 0.1 to 1.2 mg kg$^{-1}$ fresh weight, except for conventional spinach, which exhibited a higher value of 8.0 mg kg$^{-1}$ fresh weight.

In a study examining the effects of various fertilization methods on spinach growth and $NO_3^-$ content, *Vico et al. (2020)* found intriguing results. They found that the spinach yield ranged between 32.2 and 50.5 t ha$^{-1}$, with the highest yields observed in plots treated with certain fertilizers. Furthermore, spinach fertilized with fresh organic amendments, such as digested sewage sludge and composted cow manure, exhibited the highest $NO_3^-$ concentrations in leaves, ranging from 1,906–280 mg kg$^{-1}$ fresh weight. Interestingly, they noted that organic-based fertilizers resulted in similar yields to conventionally managed fields, suggesting the potential for organic practices to maintain productivity while potentially influencing $NO_3^-$ accumulation. Organic-based fertilizers may enhance soil health and microbial activity, leading to improved nutrient availability and uptake by plants, hence resulting in comparable yields to conventionally managed fields (*Verma, Pramanik & Bhaduri, 2020*). However, the higher $NO_3^-$ concentrations observed in organically fertilized spinach could be attributed to the slower release of nutrients from organic sources, which may result in prolonged exposure to $NO_3^-$ accumulation in plant tissues (*Zandvakili et al., 2019*). This trade-off highlights the importance of balancing agronomic practices to optimize yield while minimizing $NO_3^-$ accumulation, ultimately ensuring both productivity and food safety.

Controlled-release fertilizers offer a potential solution by minimizing nutrient loss and promoting plant uptake (*Trenkel, 2010*). This allows for reduced fertilizer application rates while maintaining yields, and significantly reducing $NO_3^-$ accumulation (*Trenkel, 2010*). Other strategies, like combining fertilizers with dicyandiamide (DCD) or zeolite (*Elrys et al., 2021*) *or* using slow-release release fertilizers (*Wang et al., 2020*), have also shown promise in mitigating $NO_3^-$ accumulation. However, research on the effectiveness of these methods specifically in leafy vegetables, like spinach, remains limited.

Understanding the effects of agricultural practices on $NO_3^-$ accumulation in leafy vegetables is of great importance. This study investigates the effects of four different N fertilizer types (ammonium sulfate, slow-release fertilizer with DCD inhibitor, calcium nitrate, and yeast residue) applied at varying N doses (25, 50, 100, 200, 400 mg N kg$^{-1}$) under controlled greenhouse conditions. We aim to evaluate the impact of treatments on yield, nutritional status, and quality parameters, and most importantly $NO_3^-$ accumulation in spinach plants. This research is expected to provide valuable guidance for optimizing fertilizer use and agricultural practices to control $NO_3^-$ accumulation in spinach cultivation.

## MATERIALS AND METHODS

This study was conducted in the Research and Application Greenhouses of the Department of Soil Science and Plant Nutrition, Faculty of Agriculture, Çukurova University, Turkiye. Temperature, relative humidity, and light density in the greenhouse fluctuated between 25 °C and 35 °C, 70% and 85% and 25 and 28 klux, respectively, during

the experiment. The plant material used was the Matador spinach (*Spinacia oleracea* L.) variety, which is a broad-leaved variety. This spinach variety, characterized by rapid development, large and short-petioled dark green leaves with smooth texture and oval tips, exhibiting a spreading growth habit. Additionally, it boasts high productivity and cold tolerance, making it suitable for cultivation throughout Turkiye. Ideal germination and growth occur between 15 °C and 25 °C.

The soil used in the experiment was obtained from the Research and Application Fields of Agricultural Faculty. The pH of the soil was 8.50, indicating alkaline conditions. Electrical conductivity was relatively low at 0.23 mmhos/cm, indicating non saline conditions. The soil had a high calcium carbonate content (29.1%) and a moderate organic matter content (1.20%). It also contained 13.4 mg P $kg^{-1}$ soil, 375.2 mg K $kg^{-1}$ soil, 355.2 mg Mg $kg^{-1}$ soil, 1.46 mg Cu $kg^{-1}$ soil, 0.55 mg Zn $kg^{-1}$ soil, 6.43 mg Fe $kg^{-1}$ soil, and 10.37 mg Mn $kg^{-1}$ soil.

## Greenhouse experiment

The experimental layout followed a completely randomized block design with three replicates, utilizing a total of 60 pots. Four different N fertilizer types with distinct characteristics were utilized in the experiment. The fertilizer types used in the experiment were ammonium sulphate (AS), slow-release ammonium sulphate (SRAS), calcium nitrate (CN), and yeast residue (YR). Five different nitrogen (N) doses (25, 50, 100, 200, and 400 mg N $kg^{-1}$) were applied in the form of SRAS, AS, CN, and YR. AS is the fertilizer with the highest N content (21% N), which does not contain inhibitors, organic matter, or organic carbon. CN, on the other hand, has a lower N content (11.8% N) compared to AS, and lacks inhibitors, organic matter, or organic carbon. SRAS with dicyandiamide (DCD) inhibitor contains the same 21% N as AS, but it includes a DCD inhibitor, which helps slow down the release of nitrogen. The DCD prevents the conversion of ammonium to nitrate in soil. This effect is believed to be caused by DCD binding to the active sites of ammonia monooxygenase, a copper-containing metalloenzyme crucial for ammonia-oxidizing bacteria (*Amberger, 1989*). This strategy enhances nitrogen use efficiency by mitigating nitrogen losses *via* leaching and denitrification processes. Consequently, DCD application improves fertilizer efficacy while minimizing environmental concerns such as nitrate contamination of water resources and the release of greenhouse gases. SRAS does not contain organic matter or organic carbon. YR (organic nitrogen source), has the lowest N content (3% N). It contains 35% organic matter and 16% organic carbon but does not include any inhibitors.

The experiment utilized plastic pots, each containing 2 kg of soil. The pot used in the greenhouse experiment measured 18 cm in height, 20 cm in diameter, and 18 cm in depth. As a base fertilizer application, all pots received 100 mg $kg^{-1}$ phosphorus (P) in the form of $KH_2PO_4$, 125 mg $kg^{-1}$ K in the form of $KH_2PO_4$, 10 mg $kg^{-1}$ Fe in the form of Fe-EDTA, and 2.5 mg $kg^{-1}$ Zn in the form of $ZnSO_4$. The amount of sulfur in pots receiving ammonium sulfate was calculated, and $CaSO_4$ was applied to all pots to ensure equal sulfur content.

The experiment began with seeding (10 seeds per pot) on February 24th, 2021. Seeds germinated within approximately 8–10 days. After about 16 days (March 10th, 2021), seedlings were thinned to maintain six plants per pot. Throughout the growing season, pots were watered whenever needed to maintain soil moisture content close to field capacity, allowing for free drainage to occur. On April 16, 2021, 52-day-old plants were harvested, coinciding with the observation of significant differences in growth and development due to the varying N fertilizer types and increasing N application rates.

## Measurements of spinach parameters

Soil Plant Analysis Development (SPAD) measurements were taken on a fully mature young leaf priror to harvest (52nd day of the experiment, April 16, 2021). SPAD values were measured by averaging three readings taken from the midpoints of spinach leaves using a portable chlorophyll meter (SPAD-502 Minolta Camera Co., Tokyo, Japan) (*Cordeiro, Alcantara & Barranco, 1995*). Additionally, observations were conducted on five randomly chosen plants per plot for the following parameters: plant height (cm), leaf count per plant, leaf length (cm), leaf width (cm), and fresh leaf weight per plant (g). Plant height was measured as the distance from the ground to the highest point of a leaf. Leaf length was determined by measuring from the base of the petiole (leaf stalk) to the tip of the leaf blade. Leaf width was not a single measurement but the average of three widths taken across the leaf blade at 25%, 50%, and 75% of its total length.

## Analysis of leaf samples

Upon harvest, six plants from each pot were carefully divided into three equal groups for separate storage periods. Two plants were allocated to each storage period to ensure balanced representation within each group. The first portion (Day 0) was washed and dried immediately in a 48 °C oven. The second portion (Day 5) was washed, placed in polyethylene bags, and stored in a refrigerator (+4 °C) for 5 days. Similarly, the third portion (Day 10) was washed, bagged, and refrigerated (+4 °C) for 10 days. After completing the storage periods, the plant samples were dried in an oven at 70 °C for 48 h for the analysis. The selection of storage durations (Day 0, Day 5, and Day 10) was carefully considered to capture the range of potential storage periods encountered by consumers. Day 0 represents the immediate post-harvest stage, reflecting the quality and nutrient content of spinach at the time of purchase. This point is particularly relevant for consumers who purchase and consume spinach within a short period. Day 5 represents an intermediate storage period, simulating situations where spinach is stored for a few days before consumption. This duration aligns with common home storage practices and provides insights into the quality changes that may occur during this period. Day 10 represents an extended storage period, mimicking scenarios where spinach is stored for a longer duration before consumption. This duration is particularly relevant for commercial storage and transportation practices, allowing for the assessment of long-term quality changes.

The selection of these storage durations is supported by research from *Mudau et al. (2015)*, who investigated the impact of storage temperature and duration on the nutritional

quality of spinach. Their findings indicate that that the concentrations of magnesium, zinc, and iron decreased after 8 days of storage at 4 °C. Similarly, it is noted that samples stored at 4 °C exhibited significantly higher levels of carotenoids up to 6 days, while the total phenolic compounds gradually decreased. Additionally, it is mentioned that the total antioxidant activities and vitamin C content showed a similar trend, remaining stable at 4 °C but decreasing after 6 days.

The nitrate content in plant samples was measured colourimetrically using a method developed by *Cataldo et al. (1975)*. This method relies on the formation of a yellow color complex in a strongly acidic environment. The intensity of the color complex is directly proportional to the nitrate concentration. Dried and finely ground plant samples were suspended in distilled water and incubated at 45 °C for 1 h, followed by centrifugation for 15 min. A clear-colored aliquot was mixed with 5% salicylic acid in $H_2SO_4$ and allowed to stand for 20 min. Then, 2 N NaOH was added while gently stirring, and the absorbance was measured at 410 nm using a spectrophotometer relative to the reference sample. The concentration of nitrates in the sample was measured by comparing it to a standard curve created using potassium nitrate ($KNO_3$). The results are expressed as miligrams of nitrate per gram of fresh weight of the sample.

## Soil analysis

The soil used in the study was sieved through a 2 mm mesh to remove rocks and roots. Electrical conductivity (EC) and pH of soil samples were determined in 1 soil: 2.5 deionized water mixture using the method described by *Rhoades (1983)* Calcium carbonate content was measured by estimating the quantity of the $CO_2$ produced by HCl addition to the soil. Organic matter content was analysed using the Walkley–Black dichromate oxidation procedure (*Nelson & Sommers, 1982*). Available P was extracted with 0·5 M sodium bicarbonate ($NaHCO_3$) (*Olsen et al., 1954*) and determined by spectrophotometry. The available concentrations zinc (Zn), iron (Fe), manganase (Mn) and cupper (Cu) were determined after extraction with DTPA solution (*Lindsay & Norvell, 1978*). Available nitrogen potassium (K) and magnesium (Mg) contents in soil were determined from the neutral 1 mol/L ammonium acetate extracts (1:5, m/V) and measured by a flame Photometer (*Knudsen, Peterson & Pratt, 1982*).

## Statistical analysis

All measured variables were subjected to statistical analysis using the SPSS software package. A variance analysis (ANOVA) was conducted to assess the presence of statistically significant differences among the means of treatment groups. In the experiment, four N fertilizer types, five N doses, and three different storage periods were included as the factors. The effects of individual factors, as well as the effects of two-way and three-way interactions, were analyzed for the data obtained in the experiment. Following a significant ANOVA result ($p < 0.05$), a *post-hoc* test, the Least Significant Difference (LSD) test, was employed to identify specific pairwise differences between treatment means.
## RESULTS

### The effect of slow-release, chemical, and organic nitrogen fertilizer applications on some plant characteristics of spinach

The number of leaves, leaf length, leaf width, green part length and SPAD value varied significantly among different fertilizer types ($P = 0.01$) and nitrogen (N) doses ($P = 0.01$) (Table 1). Among the fertilizer types, the highest number of leaves was recorded in the calcium nitrate (CN) $\times$ 400 mg N kg$^{-1}$ treatment (58.7 leaves pot$^{-1}$), and the lowest number of leaves was obtained in yeast residue (YR)$\times$25 mg N kg$^{-1}$ treatment (32.0 leaves pot$^{-1}$). For the ammonium sulphate (AS) treatment, the leaf number increased progressively with increasing N doses. Specifically, the AS$\times$200 treatment exhibited the highest leaf number (50.0 $\pm$ 0.0 leaves pot$^{-1}$) among AS treatments, followed closely by AS$\times$50 (44.0 $\pm$ 0.8 leaves pot$^{-1}$) and AS$\times$25 (35.0 $\pm$ 0.8 leaves pot$^{-1}$) treatment. In the case of the slow-release ammonium sulphate (SRAS) treatment, leaf number also showed variability across different N doses. The highest leaf number (51.0 $\pm$ 0.0 leaves pot$^{-1}$) was obtained in the SRAS $\times$ 200 treament, followed by SRAS $\times$ 400 (45.3 $\pm$ 0.5 leaves pot$^{-1}$), SRAS $\times$ 50 (42.0 $\pm$ 0.0 leaves pot$^{-1}$) and SRAS $\times$ 25 treatment (34.0 $\pm$ 0.8 leaves pot$^{-1}$). The CN$\times$400 treatment had the highest leaf number (58.7 $\pm$ 0.5 leaves pot$^{-1}$) among CN treatments, followed by CN$\times$200 (54.0 $\pm$ 0.0 leaves pot$^{-1}$), CN$\times$100 (53.0 $\pm$ 0.8 leaves pot$^{-1}$) and CN$\times$25 (38.0 $\pm$ 0.8 leaves pot$^{-1}$) treatments. For the YR treatment, there was a notable variation in leaf number across N doses. The YR$\times$100 treatment had the highest leaf number (52.0 $\pm$ 0.8 leaves/pot), followed by YR$\times$400 (49.0 $\pm$ 0.8 leaves pot$^{-1}$), YR$\times$50 (37.0 $\pm$ 0.0 leaves pot$^{-1}$) and YR$\times$25 (32.0 $\pm$ 0.0 leaves pot$^{-1}$).

Ammonium sulphate (21% N) led to moderate increases in leaf length and width with increasing N doses (Table 1). However, similar to the number of leaves, leaf length at the highest dose of 400 mg N kg$^{-1}$ was lower compared to 200 mg N kg$^{-1}$ N dose. The leaf length in SRAS treatments significantly increased with increasing N doses upto 200 mg N kg$^{-1}$ N dose and remained constant at the highest N dose. The difference in leaf length response between AS and SRAS fertilizers suggest that the slow-release mechanism might have contributed to better nutrient uptake even at the highest N application dose. Calcium nitrate, with a N content of 11.8%, also showed a trend of increasing leaf length and width with higher N doses. The increase in leaf length and width was relatively consistent across all doses, indicating a steady response to N supplementation. Yeast residue, with a lower N content of 3% but a higher organic matter content of 35% and organic carbon content of 16%, resulted in variable effects on leaf length and width. While lower N doses showed smaller leaf lengths compared to other fertilizers, the highest N dose led to the longest (10.3 $\pm$ 0.5 cm) and widest (5.3 $\pm$ 0.2 cm) leaves observed in the experiment.

The ANOVA revealed a significant effect of both fertilizer type and N dose on plant height ($P = 0.01$). There was also a significant interaction effect between fertilizer type and N dose ($P = 0.01$). The highest dose of YR (YR$\times$400) resulted in the tallest plants overall (23.7 cm), while lower YR doses had minimal impact (Table 1). The CN treatments generally produced taller plants compared to some AS or SRAS treatments, with CN$\times$400 (20.5 cm) being the tallest among CN groups. AS and SRAS showed the most variation,

**Table 1 Impact of fertilizer type and nitrogen doses on leaf properties and SPAD values in spinach.** Each cell in the table contains mean values ± standard error of mean for the leaf parameters and SPAD readings along with the letters for statistical analysis.

| | Leaf number | Leaf length | Leaf width | Plant length | SPAD |
|---|---|---|---|---|---|
| | Leaf/pot | cm | | | |
| AS×25 | 35.0 ± 0.8 l** | 7.1 ± 0.4 fg | 3.4 ± 0.1 fgh | 14.3 ± 0.2 f | 37.3 ± 0.6 efg |
| AS×50 | 44.0 ± 0.8 h | 7.5 ± 0.4 c-f | 4.0 ± 0.2 cd | 17.2 ± 0.4 d | 26.0 ± 0.7 h |
| AS×100 | 41.0 ± 0.8 ij | 8.7 ± 0.6 bc | 3.6 ± 0.1 ef | 17.3 ± 0.2 d | 32.9 ± 0.6 fgh |
| AS×200 | 50.0 ± 0.0 ef | 8.2 ± 0.6 b-f | 4.0 ± 0.0 cd | 20.8 ± 0.6 b | 55.3 ± 1.4 ab |
| AS×400 | 44.3 ± 0.5 h | 7.7 ± 0.7 c-f | 3.6 ± 0.0 ef | 18.1 ± 0.3 d | 58.1 ± 0.8 ab |
| CN×25 | 38.0 ± 0.8 k | 6.2 ± 0.2 gh | 3.1 ± 0.1 h | 14.2 ± 0.1 f | 32.5 ± 0.7 gh |
| CN×50 | 47.0 ± 0.8 g | 7.3 ± 0.6 efg | 3.4 ± 0.3 e-h | 14.2 ± 0.2 f | 27.2 ± 0.4 h |
| CN×100 | 53.0 ± 0.8 bc | 7.5 ± 0.0 c-f | 3.3 ± 0.2 fgh | 17.2 ± 0.2 d | 43.2 ± 0.8 cde |
| CN×200 | 54.0 ± 0.0 b | 7.5 ± 0.4 c-f | 4.0 ± 0.1 cd | 17.5 ± 0.4 d | 56.6 ± 0.5 ab |
| CN×400 | 58.7 ± 0.5 a | 8.3 ± 0.5 b-e | 4.4 ± 0.1 b | 20.5 ± 0.4 b | 51.5 ± 0.6 bc |
| YR×25 | 32.0 ± 0.0 m | 5.7 ± 0.2 h | 3.1 ± 0.1 gh | 13.7 ± 0.5 f | 34.8 ± 1.1 e-h |
| YR×50 | 37.0 ± 0.0 k | 7.2 ± 0.6 efg | 3.8 ± 0.1 de | 14.0 ± 0.4 f | 38.7 ± 1.4efg |
| YR×100 | 52.0 ± 0.8 cd | 7.6 ± 0.4 c-f | 4.2 ± 0.2 bcd | 17.2 ± 0.2 d | 41.9 ± 0.8 def |
| YR×200 | 40.0 ± 0.8 j | 7.2 ± 0.6 efg | 4.2 ± 0.2 bc | 17.7 ± 0.3 d | 50.4 ± 0.7 bcd |
| YR×400 | 49.0 ± 0.8 f | 10.3 ± 0.5 a | 5.3 ± 0.2 a | 23.7 ± 0.8 a | 50.1 ± 0.8 bcd |
| SRAS × 25 | 34.0 ± 0.8 l | 7.3 ± 0.5 d-g | 3.4 ± 0.1 e-h | 15.3 ± 0.5 e | 42.1 ± 0.7 def |
| SRAS × 50 | 42.0 ± 0.0 i | 7.0 ± 0. fg | 3.5 ± 0.2 efg | 15.4 ± 0.4 e | 34.2 ± 0.5 e-h |
| SRAS × 100 | 37.0 ± 0.8 k | 8.5 ± 0.4 bcd | 4.2 ± 0.2 bc | 19.3 ± 0.2 c | 36.9 ± 0.4 efg |
| SRAS × 200 | 51.0 ± 0.0 de | 9.0 ± 0.4 b | 4.2 ± 0.2 bcd | 20.4 ± 0.3 b | 52.8 ± 1.3 b |
| SRAS × 400 | 45.3 ± 0.5 h | 9.0 ± 0.8 b | 4.1 ± 0.1 bcd | 20.4 ± 0.8 b | 62.0 ± 5.8 a |
| Fertilizer (F) $P$:LSD value | 0.01:0.57 | 0.01:0.47 | 0.01:0.15 | 0.01:0.41 | 0.16:3.58 |
| Nitrogen dose (ND) $P$:LSD value | 0.01:0.64 | 0.01:0.52 | 0.01:0.17 | 0.01:0.46 | 0.01:4.01 |
| F × ND $P$:LSD value | 0.01:1.28 | 0.01:1. | 0.01:0.35 | 0.01:0.91 | 0.01:8.01 |

**Note:**

** Values with the same letter in the table do not significantly differ from each other ($p < 0.05$), AS, Ammonium Sulphate; CN, Calcium Nitrate; YR, Yeast Residue; SRAS, Slow Release Ammonium Sulphate; Nitrogen Doses (mg/kg).

with AS×200 achieving a height comparable to the tallest CN treatment, but other AS and SRAS doses resulting in shorter plants.

SPAD values, an indicator of chlorophyll content and leaf health, showed significant variability across different fertilizer types and N doses. In AS treatments, SPAD values increased from 37.3 ± 0.6 (AS×25) to 58.1 ± 0.8 (AS×400), indicating a 35.8% increase in chlorophyll content with increasing AS doses. Similar to AS treatments, SPAD values ranged from 34.2 ± 0.5 (SRAS×50) to 62.0 ± 5.8 (SRAS×400), showing a substantial increase in chlorophyll content with increasing SRAS doses. SPAD values in CN treatments ranged from 27.2 ± 0.4 (CN×50) to 58.7 ± 0.5 (CN×400), suggesting a positive

**Table 2 The effects of different N sources and doses on fresh weight of spinach plants during different storage days.** Each cell in the table contains mean ± standard error of mean values for the fresh weights of spinach plants in each pot along with the letters for statistical analysis.

| Period | N dose | Ammonium sulphate | Slow release ammonium sulphate | Calcium nitrate | Yeast residue |
|--------|--------|-------------------|--------------------------------|-----------------|---------------|
|        | mg/kg  | g/pot |  |  |  |
| Day 0  | 25     | 8.2 ± 1.7 j-s* | 5.6 ± 0.5 p-s | 8.0 ± 1.3 j-s | 7.0 ± 1.1 l-s |
|        | 50     | 10.5 ± 1.4 d-p | 8.1 ± 0.7 j-s | 9.2 ± 0.8 h-s | 7.8 ± 0.3 fk-s |
|        | 100    | 10.9 ± 1.3 d-o | 9.6 ± 0.2 g-r | 12.9 ± 2.0 a-k | 12.0 ± 2.4 a-l |
|        | 200    | 15.3 ± 1.0 a-d | 14.8 ± 2.3 a-g | 16.2 ± 2.2 abc | 13.3 ± 0.7 a-j |
|        | 400    | 11.5 ± 1.0 b-n | 14.4 ± 0.7 a-h | 11.8 ± 0.8 a-m | 16.6 ± 0.5 ab |
| Day 5  | 25     | 7.8 ± 1.6 k-s | 5.9 ± 0.9 o-s | 7.4 ± 0.9 l-s | 4.7 ± 0.3 rs |
|        | 50     | 8.7 ± 2.3 I-s | 8.3 ± 0.7 j-s | 8.9 ± 0.3 I-s | 7.3 ± 0.9 s |
|        | 100    | 11.9 ± 3.1 a-m | 12.2 ± 2.5 a-l | 12.1 ± 1.5 a-l | 9.0 ± 1.1 i-s |
|        | 200    | 14.9 ± 0.9 a-f | 11.2 ± 1.1 c-n | 9.9 ± 1.3 f-r | 10.9 ± 0.4 d-p |
|        | 400    | 13.1 ± 1.4 a-j | 13.9 ± 0.3 a-i | 15.2 ± 3.5 a-e | 14.3 ± 1.3 a-h |
| Day 10 | 25     | 4.8 ± 2.2 qrs | 4.1 ± 1.2 s | 6.3 ± 0.6 n-s | 4.1 ± 0.6 s |
|        | 50     | 5.6 ± 1.0 p-s | 6.6 ± 1.2 m-s | 8.4 ± 2.4 j-s | 6.2 ± 0.7 n-s |
|        | 100    | 8.4 ± 2.4 j-s | 5.2 ± 0.7 qrs | 9.9 ± 1.9 f-r | 8.0 ± 0.5 j-s |
|        | 200    | 10.1 ± 1.9 e-p | 10.9 ± 1.4 d-p | 14.4 ± 1.1 a-h | 7.6 ± 1.6 k-s |
|        | 400    | 8.5 ± 1.7 j-s | 14.7 ± 0.4 a-g | 14.7 ± 3.0 a-g | 16.9 ± 1.7 a |
| ANOVA  | Day    | LSD = 0.94 $P$ = 0.001 |  |  |  |
|        | Fertilizer type | LSD = 1.085 $P$ = 0.054 |  |  |  |
|        | Nitrogen dose | LSD = 1.213 $P$ = 0.001 |  |  |  |
|        | Day × Fertilizer type | LSD = 1.88 $P$ = 0.080 |  |  |  |
|        | Day × Nitrogen dose | LSD = 2.112 $P$ = 0.095 |  |  |  |
|        | Fertilizer type × Nitrogen dose | LSD = 2.427 $P$ = 0.010 |  |  |  |
|        | Day × Fertilizer type × Nitrogen dose | LSD = 4.203 $P$ = 0.752 |  |  |  |

**Note:**
* Values with the same letter in the table do not significantly differ from each other ($p < 0.05$).

effect on chlorophyll content. SPAD values YR treatments ranged from 34.8 ± 1.1 (YR×25) to 50.4 ± 0.7 (YR×200), indicating a moderate increase in SPAD values with increasing YR doses.

## The effect of different fertilizer types and doses on fresh weight

Fresh weight of spinach plants at each storage period (0, 5, and 10 days) was significantly different from each other. The effect of fertilizer type on fresh weight was not statistically significant ($P$ = 0.054). Additionally, significant differences ($P$ = 0.01) were evident within each fertilizer type across the different N doses (Table 2). The effect of fertilizer type × N dose interaction on fresh weight was significant ($P$ = 0.010), while Day × fertilizer type ($P$ = 0.080), Day × N dose ($P$ = 0.095) and Day × Fertilizer Type × N Dose ($P$ = 0.752) interactions were not statistically significant.

At the begining of storage period (day 0), the AS and CN fertilizers showed an increase in fresh weights of spinach with increasing N dose up to 200 mg kg$^{-1}$, followed by a decrease at 400 mg N kg$^{-1}$ (Table 2). On Day 5, the highest yield in the AS application was

recorded at the 200 mg N kg$^{-1}$ N dose (14.9 g pot$^{-1}$), while the highest fresh spinach weights in other fertilizer types were obtained at the 400 mg N kg$^{-1}$ N dose (Table 2). A similar pattern of weight increase and decrease with increasing N dose was observed in the AS application throughout all days. Weight increased up to the 200 mg N kg$^{-1}$ N dose on all days but decreased at the 400 mg N kg$^{-1}$ N dose. In SRAS, the slow-release form of AS, weight increased with increasing N dose up to 200 mg N kg$^{-1}$ on Day 0, and there was a statistically insignificant decrease at the 400 mg N kg$^{-1}$ N dose. However, on Day 5 and Day 10, weight consistently increased with increasing N dose. In YR, spinach fresh yield consistently increased with increasing N dose. The increase became more pronounced when increasing the N dose from 200 to 400 mg N kg$^{-1}$ every 3 days. The weight, initially 13.3 g on the Day 0, increased to 16.6, 10.9 g pot$^{-1}$ on Day 5 increased to 14.3 g pot$^{-1}$, and the weight of 7.6 g pot$^{-1}$ on Day 10 increased to 16.9 g pot$^{-1}$ at the 400 mg N kg$^{-1}$ N dose. In the CN application, a similar trend to AS was observed on Day 0, while on Day 5 and Day 10, an increase in fresh spinach yield was observed with increasing N dose.

## The effect of different fertilizer types and doses on $NO_3^-$ content of spinach plants

The effect of increasing doses of slow-release, chemical, and yeast residue fertilizers, as well as their different doses, on $NO_3^-$ content of spinach plant was determined by drying and analyzing both the leaf blade and petiole together. The average $NO_3^-$ content in spinach plants during Day 0, 5, and 10 of the storage period are shown in Table 3. Variance analysis revealed that nitrogen fertilizer type, dose, and storage day as well as their interactions significantly influenced the nitrate concentration of spinach plants. All fertilizer types exhibited a decrease in $NO_3^-$ concentration of spinach plants between Day 0 and Day 10. Nitrate concentration generally decreased over the storage period for all nitrogen fertilizer types and doses. For example, the $NO_3^-$content in spinach plants treated with 400 mg/kg AS on Day 0 (2,668 mg $NO_3^-$ kg$^{-1}$) declined to 2,130 mg $NO_3^-$ kg$^{-1}$ by Day 10.

The $NO_3^-$ content in the YR fertilizer treatments varied across different days. On Day 0, the $NO_3^-$ concentrations were higher compared to other fertilizers, with values ranging from 1,431 to 1,812 mg $NO_3^-$ kg$^{-1}$ for YR × 25 and YR × 400, respectively. However, as time progressed, there was a general decrease in $NO_3^-$ content. On Day 5, the $NO_3^-$ concentrations increased, ranging from 951 mg $NO_3^-$ kg$^{-1}$ to 1,882 mg $NO_3^-$ kg$^{-1}$ for YR × 25 and YR × 400, respectively. Finally, by Day 10, the $NO_3^-$ concentrations further decreased, with values ranging from 377 mg $NO_3^-$ kg$^{-1}$ to 1,385 mg $NO_3^-$ kg$^{-1}$ for YR × 25 and YR × 400, respectively (Table 3). As expected, $NO_3^-$ content generally increased with higher N doses. For instance, on Day 0, the concentration in spinach treated with 400 mg/kg ammonium sulfate was almost double that of the 25 mg/kg dose (2,668 mg $NO_3^-$ kg$^{-1}$ *vs.* 850 mg $NO_3^-$ kg$^{-1}$) (Table 3).

## DISCUSSION

### Morphological characteristics of spinach leaves

This study demonstrated a positive impact of N fertilization on spinach growth parameters like leaf size and plant height. All fertilizers increased leaf length and width with higher N

**Table 3 The effects of different N sources and doses on nitrate concentration of spinach plants during different storage days.** Each cell in the table contains mean ± standard error of mean values for the nitrate contents of spinach plants in each pot along with the letters for statistical analysis.

| Period | N dose | Ammonium sulphate | Slow release ammonium sulphate | Calcium nitrate | Yeast residue |
|---|---|---|---|---|---|
| | mg/kg | mg/kg | | | |
| Day 0 | 25 | 850 ± 42 opg* | 945 ± 41 gh | 1,203 ± 17 jkl | 1,431 ± 126 hi |
| | 50 | 882 ± 53 m-q | 961 ± 30 gh | 1,234 ± 22 ijk | 1,436 ± 66 hi |
| | 100 | 960 ± 23 mno | 961 ± 32 gh | 1,258 ± 86 ijk | 1,700 ± 183 fg |
| | 200 | 1,390 ± 33 ij | 1,029 ± 61 fgh | 1,868 ± 210 f | 1,717 ± 161 fg |
| | 400 | 2,668 ± 12 d | 2,134 ± 175 c | 4,395 ± 488 e | 1,812 ± 159 f |
| Day 5 | 25 | 831 ± 60 opg | 860 ± 46 n-q | 1,066 ± 9 k-n | 951 ± 12 mno |
| | 50 | 863 ± 31 n-q | 904 ± 17 m-q | 1,189 ± 82 jkl | 1,080 ± 92 klm |
| | 100 | 931 ± 116 mno | 929 ± 79 mno | 1,206 ± 60 jkl | 1,284 ± 134 ij |
| | 200 | 1,365 ± 82 ij | 987 ± 68 mno | 1,605 ± 34 gh | 1,428 ± 18 hi |
| | 400 | 2,544 ± 288 d | 2,082 ± 107 e | 3,418 ± 220 b | 1,882 ± 118 f |
| Day 10 | 25 | 405 ± 64 tu | 369 ± 54 u | 462 ± 37 stu | 377 ± 16 u |
| | 50 | 429 ± 13 tu | 586 ± 21 rst | 486 ± 5 stu | 394 ± 8 tu |
| | 100 | 488 ± 14 stu | 707 ± 40 qr | 587 ± 6 rst | 640 ± 15 rs |
| | 200 | 915 ± 73 m-p | 715 ± 7 pqr | 981 ± 52 mno | 1,026 ± 45 l-o |
| | 400 | 2,130 ± 41 e | 2,069 ± 47 e | 2,976 ± 136 c | 1,385 ± 17 ij |
| ANOVA | Day | LSD = 38.968 $P$ = 0.001 | | | |
| | Fertilizer type | LSD = 44.996 $P$ = 0.001 | | | |
| | Nitrogen dose | LSD = 50.307 $P$ = 0.001 | | | |
| | Day × Fertilizer type | LSD = 77.936 $P$ = 0.001 | | | |
| | Day × Nitrogen dose | LSD = 87.135 $P$ = 0.048 | | | |
| | Fertilizer type × Nitrogen dose | LSD = 100.615 $P$ = 0.001 | | | |
| | Day × Fertilizer type × Nitrogen dose | LSD = 174.27 $P$ = 0.001 | | | |

**Note:**
* Values with the same letter in the table do not significantly differ from each other ($p < 0.05$).

doses, with AS and CN showing the most consistent response. This aligns with the findings of *Özenç & Şenlikoğlu (2017)* who reported a 12% increase in spinach leaf number with higher N doses, and *Purquerio et al. (2007)* who observed larger leaf area in arugula plants with higher N doses. These findings highlight the importance of both N content and organic matter characteristics when selecting fertilizers for for leafy crops.

Plant length, a crucial morphological parameter indicating plant vigor and growth of plants significantly increased with higher N doses across fertilizer types. The most pronounced increase was observed for CN treatment (44% increase at the highest N dose), possibly due to calcium's role in cell wall structure enhancement, as reported by *Kacar & Katkat (2015)*. Additionally, reduced sodium uptake associated with CN fertilization, as observed by *Ebert et al. (2002)*, may contribute to improved plant growth. This aligns with findings from *Thapa et al. (2021)*, *Zaman et al. (2018)* and *Shormin & Kibria (2018)* who reported increased plant height and leaf number with high N application. The observed increase in plant growth parameters can likely be attributed to enhanced photosynthetic activity due to increased N availability (*Kubar et al., 2022*).

Chlorophyll content indicated by SPAD values, a generally increased with higher N doses for AS, SRAS, and CN treatments, indicating improved chlorophyll systhesis leaf health. The YR treatment resulted in moderate increases in SPAD values, suggesting a less pronounced effect compared to AS, CN, and SRAS. The positive relationship between plant N nutrition and SPAD values observed in this study aligns with established knowledge (*Esfahani et al., 2008*; *Hou et al. 2021*). As reported by *Porter & Evans (1998)*, increasing N concentration in leaves, associated with higher N application, enhances the intensity of light utilized during photosynthesis. However, while we did not directly measure photosynthesis, the positive relationship between SPAD values and N dose in our study suggests potentially improved photosynthetic activity, particularly for AS and SRAS treatments with the highest SPAD values. The observed variability in SPAD values across N fertilizer types suggests that fertilizer type, beyond just N content, may influence chlorophyll content and potentially photosynthetic performance. Future studies directly measuring photosynthesis are needed to confirm this hypothesis. Although *Han et al. (2023)* highlighted the complexity of $NO_3^-$ stress on spinach leaves, our results generally support the positive relationship between N dose and SPAD values. Future studies should directly measure photosynthesis to confirm these findings and explore potential strategies for mitigating $NO_3^-$ stress to ensure food safety.

## Fresh weights of spinach plants during various storage periods

Numerous studies have highlighted the positive effects of N application and various N fertilizer types on plant growth and yield (*Albayrak & Çamaş, 2006*; *Tekeli & Daşgan, 2013*). Nitrogen serves a pivotal role in stimulating vegetative growth, enhancing leaf development, and improving overall plant health. Furthermore, N fertilization has been shown to enhance photosynthesis, leading to increased biomass production and improved crop yields (*Züst & Agrawal, 2016*). However, recent findings by *Han et al. (2023)* suggest a more nuanced relationship between $NO_3^-$ levels and plant growth. Their study revealed that excessive $NO_3^-$ concentrations could significantly reduce plant biomass, indicating negative effects on growth. Similarly, our observations indicated a decrease in spinach plant fresh weights at the highest N doses (400 mg kg$^{-1}$) for all N fertilizer types, highlighting the importance of optimizing fertilization to avoid adverse effects on growth and yield.

While N application can enhance plant growth and yield, increasing it beyond a certain point can be counterproductive (*The, Snyder & Tegeder, 2021*). Our study, along with findings by *Han et al. (2023)*, underscores the importance of optimizing N fertilization strategies. This is crucial to maximize plant growth and yield while minimizing negative environmental consequences, such as N emissions (*Guo, Liu & He, 2022*; *Menegat, Ledo & Tirado, 2022*). Our study also revealed differences in yield response among different N fertilizer types. YR exhibited a continuous yield increase, indicating a potentially slower N release compared to AS and CN, which showed a decrease at the highest N dose. SRAS shows a similar increase as AS up to moderate N dose (200 mg N kg$^{-1}$), but without the sharp decline observed with AS and CN at the highest dose (400 mg N kg$^{-1}$). This suggests a slower release profile for SRAS compared to AS, but without the negative impact of

excess N. In general, the AS generally outperformed SRAS in yield across all N doses on Day 0. This suggests a quicker availability of N from AS for early plant growth.

The optimal N dose may vary depending on fertilizer type. AS and CN may benefit most from a 200 mg N kg$^{-1}$ dose on Day 0, while YR and SRAS might require higher doses for maximum yield. The organic nature of YR and the controlled release mechanism of SRAS likely explain their delayed response. Their full effects might be realized later in the growth cycle.

## Nitrate content of spinach plants during various storage periods

Increasing N application doses generally led to higher initial $NO_3^-$ concentrations across all fertilizer types and days. Our findings are consistent with *Liu et al. (2006)*, who observed a dramatic increase in leaf $NO_3^-$ concentration at the highest N application rate (240 mg N kg$^{-1}$ soil), reaching 708 mg $NO_3^-$ kg$^{-1}$. Among inorganic fertilizers (AS, SRAS, CN), SRAS generally resulted in lower peak $NO_3^-$ concentrations compared to AS at equivalent application rates. This slower release and uptake pattern may contributeto steadier supply of N throughout the growth period, potentially beneficial for plant growth. The magnitude of $NO_3^-$ concentration increase varied depending on the fertilizer type. CN exhibited the most significant increase compared to AS and SRAS, highlighting the influence of fertilizer choice on $NO_3^-$ accumulation. Our findings align with *Inal & Tarakcioglu (2001)*, who stated that ammonia-based nitrogen fertilizers are less readily absorbed by most plants compared to nitrate-based fertilizers. This characteristic can be advantageous in reducing the risk of excessive nitrate uptake within crops. For instance, the lowest $NO_3^-$ content on Days 0 and 5 was observed in AS × 25 (850 and 831 mg $NO_3^-$ kg$^{-1}$, respectively) treatments, as well as SRAS × 25 (369 mg $NO_3^-$ kg$^{-1}$) and YR (377 mg $NO_3^-$ kg$^{-1}$) treatments on Day 10 (Table 2). Given the significance of spinach and other leafy green vegetables in human nutrition, it is crucial to maintain $NO_3^-$ concentration below the recommended levels to ensure the consumer safety (*Vico et al., 2020*). The European Union has established a safe limit for $NO_3^-$ levels in spinach at less than 3,500 mg $NO_3^-$ kg$^{-1}$ of fresh weight, with even lower limits for infants and young children (Regulation No 1258/2011 of 2 December 2011; *EFSA (European Food Safety Authority), 2008*). While most of our treatments remained within safe limits, the $NO_3^-$ concentration in spinach exceeded or closely approached the safe limit of 3,500 mg $NO_3^-$ kg$^{-1}$ fresh weight with CN applications at 400 mg N kg$^{-1}$ on Days 1 and 5. In line with previous research by *Vico et al. (2020)*, who reported $NO_3^-$ concentrations in spinach plants typically below established safety thresholds, most of our treatments resulted in safe $NO_3^-$ levels. Their experiment, utilizing a normalized N application rate of 150 kg ha$^{-1}$, evaluated eight different fertilizing scenarios, including inorganic NPK fertilizers, digestates, biosolids, and organic amendments like composts and vermicomposts. The resulting $NO_3^-$ levels in their spinach leaves ranged between 280 and 1,906 mg $NO_3^-$ kg$^{-1}$ fresh weight. The lower $NO_3^-$ content observed in our SRAS treatments compared to the corresponding AS treatment suggests a potential benefit of slow-release fertilizers. Similar to SRAS, slow-release CN fertilizer applications might limit the plant's N uptake rate and contribute to a decrease in accumulated $NO_3^-$.

Comparing $NO_3^-$ concentrations between Day 0 and Day 5 revealed varying trends across different N fertilizer types. The percent changes for AS ranged from −1.80% to −4.60%, indicating a consistent decrease in $NO_3^-$ concentration across all doses. SRAS showed decreases ranging from −2.44% to −8.99%, with the highest decrease observed at the lowest dose. The CN exhibited notable decreases ranging from −3.64% to −22.23%, with the largest decrease observed at the highest dose. In contrast, the YR displayed a mix of trends, with percent changes ranging from −33.61% to 3.86%. While most doses showed decreases, the highest dose experienced a slight increase in $NO_3^-$ concentration (Table 3).

The changes in $NO_3^-$ concentration in spinach plants were significant between Day 0 and Day 10 across different fertilizer types and N doses. For instance, with AS fertilizer, there was a decrease in $NO_3^-$ concentration ranging from approximately 52.94% to 20.18% as the N dose increased from 25 to 400 mg N $kg^{-1}$. Similarly, the SRAS showed a reduction in $NO_3^-$ concentration by approximately 53.02% to 18.30% across the same range of N doses. The CN exhibited a decrease in $NO_3^-$ concentration ranging from about 61.62% to 39.59%, while the YR displayed a wider range of decreases, from approximately 71.89% to 34.51% across the various N doses (Table 3). *Tamme et al. (2010)* reported that during storage of leafy vegetables (such as spinach) at room temperature, $NO_3^-$ levels in the vegetables decreased significantly by an average of 87.4% from the third day. On the other hand, nitrite levels in the leaves increase dramatically, ranging from 1,857 to 3,617 mg $kg^{-1}$. *Chung, Chou & Hwang (2004)* stated that when leafy vegetables were stored in the refrigerator, there were no significant changes in $NO_3^-$ and nitrite levels by the 7th day. In this study, while the $NO_3^-$ content in spinach leaves did not change significantly after the first 5 days, $NO_3^-$ levels decreased significantly at the end of the 10th day in all applied N fertilizer types and doses.

The type and amount of N fertilizer applied to soil are the key factors influencing $NO_3^-$ levels in vegetables. *Chohura & Kolota (2009)* found that AS fertilizer led to lower $NO_3^-$ concentrations in vegetables compared to other fertilizers. In line withthe findings of *Chohura & Kolota (2009)*, other studies revealed that fertilizers applied in the $NO_3^-$ form resulted in higher $NO_3^-$ concentrations compared to those applied in the $NH_4^+$ form. Slow-release fertilizers, with the effect of nitrification inhibitors, retain N in the soil for a longer period in the $NH_4^+$ form, delaying $NO_3^-$ formation and minimizing $NO_3^-$ accumulation in plant leaves (*Amberger, 1989*; *Inal & Tarakcioglu, 2001*). *Krezel & Kolota (2014)* did not report a significant impact of fertilizer type on $NO_3^-$ levels in spinach. The lowest $NO_3^-$ content was recorded in plants fertilized with AS (295.17 mg $NO_3^-$ $kg^{-1}$ fresh weight), while the highest was found in those given ammonium nitrate (354.02 mg $NO_3^-$ $kg^{-1}$ fresh weight).

Similar to our findings, *Liu et al. (2014)* showed that the lettuce grown without fertilizer application exhibited the lowest $NO_3^-$ concentration (1,391 mg $kg^{-1}$), while the highest content was observed in lettuce treated with inorganic fertilizers. Nitrate concentrations in lettuce treated with inorganic and organic fertilizers ranged from 5,000 to 6,100 (mg $NO_3^-$ $kg^{-1}$) and 4,300 to 5,200 (mg $NO_3^-$ $kg^{-1}$), respectively. The reseachers indiecated that the addition of liquid fertilizers resulted in a further 4–10% reduction in $NO_3^-$ concentration compared to lettuce treated solely with organic fertilizers. *Güneş (2021)* showed that the

application of AS and nitrification inhibitors DCD and DMPP under field conditions resulted in lower $NO_3^-$ concentrations in spinach plants. Slow released fertilizers are nitrification inhibitor fertilizers that inhibit the activity of nitrifying bacteria responsible for converting ammonium to nitrate, thereby allowing nitrogen to remain in the ammonium form for 4–8 weeks depending on soil conditions (*Scheffer & Bartels, 1998*). Similar to *Güneş (2021)*, *Montemurro et al. (2008)* reported that the application of the nitrification inhibitor DCD fertilizer reduced $NO_3^-$ concentration in lettuce by 24% compared to urea application.

Organic N sources like YR, consistently maintained lower $NO_3^-$ levels in spinach plants, throughout the experiment, potentially minimizing the risk of excess $NO_3^-$ accumulation compared to inorganic fertilizers. The slower release of N from organic source, which may not be fully utilized by the plants during the experimental period, could contribute to this observation. This is because organic fertilizers typically do not provide N in a readily available form (*Herencia et al., 2011*). Therefore, $NO_3^-$ accumulation in edible part of crops is usually lower in organically grown crops than in conventionally grown crops (*Pavlou, Ehaliotis & Kavvadias, 2007*). Supporting this finding, *Liu et al. (2014)* showed that lettuce grown under organic fertilizer (200 kg N ha$^{-1}$) accumulated 14 to 19% less $NO_3^-$ compared to mineral nitrogen fertilizer at (200 kg N ha$^{-1}$ as $NH_4NO_3$).

## CONCLUSIONS

This study investigated the impact of nitrogen fertilizer type and application rate on the growth, nitrate content, and storage quality of spinach plants. The findings highlight the importance of tailoring nitrogen fertilization strategies to achieve desired outcomes. For producers seeking rapid early growth and potentially higher yields, ammonium sulphate (AS) emerged as a viable option at moderate doses (200 mg N kg$^{-1}$). However, the observed decrease in growth at the highest dose (400 mg N kg$^{-1}$) underscores the importance of careful monitoring to avoid exceeding safe nitrate limits set by regulatory bodies.

Slow-release ammonium sulphate (SRAS) presented a more balanced approach. While promoting sustained growth comparable to AS at moderate doses, SRAS resulted in generally lower peak nitrate concentrations, potentially reducing the risk of exceeding safe limits. This characteristic makes SRAS a promising option for producers seeking to optimize both yield and consumer safety. Yeast residue (YR) emerged as a viable option for organic spinach production. Despite its lower nitrogen content compared to other fertilizers, YR still promoted plant growth, albeit at a slower rate. Importantly, YR consistently maintained the lowest nitrate levels throughout the experiment, offering a potential solution for organic producers concerned about nitrate accumulation.

While the study provides valuable insights into the effects of different nitrogen fertilizers on spinach growth and nitrate content. However, it is essential to acknowledge some limitations that may affect the generalizability of the findings. The study was conducted under controlled greenhouse conditions. Real-world field conditions can vary significantly in terms of temperature, light intensity, rainfall patterns, and soil properties. These factors can influence plant growth, nutrient uptake, and nitrate accumulation.

Therefore, the observed responses in the greenhouse may not translate directly to outdoor settings. In addition, the study utilized a single spinach cultivar, Matador, a broad-leaved variety. Different spinach cultivars exhibit varying responses to fertilizer application due to inherent genetic differences. Evaluating a wider range of cultivars would provide a more comprehensive understanding of fertilizer effects on spinach production. By acknowledging these limitations, we emphasize the need for further research under field conditions, using diverse spinach varieties, and considering the influence of interacting environmental factors. This broader perspective will contribute to the development of more robust and generalizable recommendations for sustainable spinach production practices.

### Funding
This study was supported by the Çukurova University Research Projects Unit. Project (No. FYL-2021-13962). The funders had no role in study design, data collection and analysis, decision to publish, or preparation of the manuscript.

### Grant Disclosures
The following grant information was disclosed by the authors:
Çukurova University Research Projects: FYL-2021-13962.

### Competing Interests
The authors declare that they have no competing interests.

### Author Contributions
- Kemal Yalçın Gülüt conceived and designed the experiments, performed the experiments, analyzed the data, prepared figures and/or tables, authored or reviewed drafts of the article, and approved the final draft.
- Gamze Güleç Şentürk conceived and designed the experiments, performed the experiments, prepared figures and/or tables, and approved the final draft.

### Data Availability
The raw data is available in the Supplemental File.

### Supplemental Information
Supplemental information for this article can be found online at http://dx.doi.org/10.7717/peerj.17726#supplemental-information.

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
