# Peer review of "Impact of nitrogen fertilizer type and application rate on growth, nitrate accumulation, and postharvest quality of spinach"

_PeerJ, doi:10.7717/peerj.17726_

## Round 0.1 · original submission · Major Revisions

The manuscript has undergone review by two experts in the field, and both have recommended significant revisions. Please ensure you carefully address their suggestions to enhance the quality of your manuscript.

**Language Note:** The review process has identified that the English language must be improved. PeerJ can provide language editing services - please contact us at [email protected] for pricing (be sure to provide your manuscript number and title). Alternatively, you should make your own arrangements to improve the language quality and provide details in your response letter. – PeerJ Staff

Reviewer 1 ·

Basic reporting

The manuscript needs minor changes

Experimental design

- The study demonstrates an acceptable experimental design.
- A completely randomized block design with three replicates and a total of 60 pots was used.
- Five nitrogen application rates (25, 50, 100, 200, and 400 mg kg-1) were tested for each nitrogen source.
- Each pot contained 2 kg of soil and initially had 10 spinach seeds. After germination, seedlings were thinned to 6 plants per pot.

Validity of the findings

The findings describe the effects of different nitrogen sources on spinach growth and development.

Additional comments

I reviewed the paper titled "Impact of Nitrogen Source and Application Rate on Growth, Nitrate Accumulation, and Postharvest Quality of Spinach". The study effectively investigates the key aspects of the research on nitrogen source and application for spinach cultivation.
-Comments and Suggestions for Authors
Abstract
- The abstract is well-written and informative.
- Consider mentioning the specific storage duration investigated in the study (e.g., 10 days) to provide better context for the nitrate content results
- Line 22: Consider replacing "better" with "improved" for a more formal tone.
- Line 30: "SPAD values" could be defined in parentheses for readers unfamiliar with the term (e.g., SPAD values (…..)).
Introduction
- Strengthen the connection between yield and nitrate: While you mention Vico et al. (2020) finding similar yields with organic amendments and conventional practices despite higher nitrate in organically fertilized spinach, you could delve deeper into the potential trade-offs between yield and nitrate content.
- Consider mentioning limitations of existing research: Briefly acknowledge the limited research on slow-release fertilizers' impact on nitrate accumulation in leafy vegetables.
Materials and methods
- This section provides a comprehensive overview of the experimental methods used. It allows readers to understand how the experiment was conducted and evaluate the validity of the results.
- Line 185: remove "Methods"
Results
- The study only investigated a single spinach cultivar. Results may not be applicable to other varieties.
- The study examined nitrate content during storage, but it's not clear if the storage conditions typically encountered by consumers were mimicked.
Discussion

- While the discussion mentions various findings, it would benefit from a more explicit connection between each result and the relevant previous research or explanation.
- The current structure jumps between discussing different aspects of the results (growth, chlorophyll, fresh weight, nitrate) multiple times. Consider grouping related results for a more focused discussion.
- Where relevant, consider mentioning the magnitude of the observed effects (e.g., percentage increase in plant height) to strengthen the impact of the findings.
- Briefly acknowledging any limitations of the study (e.g., specific conditions that may not be generalizable) can add transparency and nuance to the discussion.

Reviewer 2 ·

Basic reporting

The paper "Impact of Nitrogen Source and Application Rate on Growth, Nitrate Accumulation, and Postharvest Quality of Spinach" is dealing with an important topic which is how to manage nitrogen fertilizers to be available to plants and environment friendly for crop with high nitrogen requirement.
The paper needs an effort in English editing to present the work in more professional version. In general, the introduction and especially the discussion are too long and they need a throughout revision to be more concise and relevant. Some part could be removed and other could be combined and shortened.

Experimental design

The experiment section requires throughout revision, where providing more details is necessary for the the understanding and the reproducibility of the work. The description of the work, material, methods and conducting experiment should well organized in different section without redundancy to avoid the confusion of the reader. It would be better if the description of experiment conduction is presented in a chronological way and the different factors included in the experiment are highlighted.

Validity of the findings

'no comment'

Additional comments

Please check attached document, where you will find specific comments.

Annotated reviews are not available for download in order to protect the identity of reviewers who chose to remain anonymous.

---

## Round 0.2 · accepted · Accept

The revised version of the manuscript has been reviewed by two reviewers. Both have acknowledged the improvements and recommended its acceptance.

Reviewer 1 ·

Basic reporting

no comment

Experimental design

no comment

Validity of the findings

no comment

Additional comments

The authors have made the changes I suggested in the last review. I recommend its publication in this journal.

Reviewer 2 ·

Basic reporting

The authors have revised the manuscript accoding to the recommendations and suggestions made.

Experimental design

'no comment'

Validity of the findings

'no comment'

Additional comments

'no comment'